# Exploring the Temporal Relation between Body Mass Index and Corticosteroid Metabolite Excretion in Childhood

**DOI:** 10.3390/nu12051525

**Published:** 2020-05-23

**Authors:** Britt J. Van Keulen, Conor V. Dolan, Ruth Andrew, Brian R. Walker, Hilleke E. Hulshoff Pol, Dorret I. Boomsma, Joost Rotteveel, Martijn J.J. Finken

**Affiliations:** 1Emma Children’s Hospital, Amsterdam UMC, Vrije Universiteit Amsterdam, Pediatric Endocrinology, Amsterdam, De Boelelaan 1117, 1081 HV Amsterdam, The Netherlands; j.rotteveel@amsterdamumc.nl (J.R.); m.finken@amsterdamumc.nl (M.J.J.F.); 2Netherlands Twin Register, Department of Biological Psychology, Vrije Universiteit Amsterdam, Van der Boechorststraat 7-9, 1081 BT, Amsterdam, The Netherlands; c.v.dolan@vu.nl (C.V.D.); di.boomsma@vu.nl (D.I.B.); 3Centre for Cardiovascular Science, University of Edinburgh, Queen’s Medical Research Institute, 47, Little France Crescent, Edinburgh EH16 4TJ, UK; Ruth.Andrew@ed.ac.uk (R.A.); Brian.Walker@newcastle.ac.uk (B.R.W.); 4Institute of Genetic Medicine, Newcastle University, Central Pkwy, Newcastle upon Tyne NE1 3BZ, UK; 5Department of Psychiatry, University Medical Center Utrecht, Brain Center, Heidelberglaan 100, 3584 CX Utrecht, The Netherlands; H.E.Hulshoff@umcutrecht.nl

**Keywords:** obesity, glucocorticoids, BMI, HPA axis

## Abstract

Childhood obesity is associated with alterations in hypothalamus–pituitary–adrenal (HPA) axis activity. However, it is unknown whether these alterations are a cause or a consequence of obesity. This study aimed to explore the temporal relationship between cortisol production and metabolism, and body mass index (BMI). This prospective follow-up study included 218 children (of whom 50% were male), born between 1995 and 1996, who were assessed at the ages of 9, 12 and 17 years. Morning urine samples were collected for assessment of cortisol metabolites by gas chromatography-tandem mass spectrometry, enabling the calculation of cortisol metabolite excretion rate and cortisol metabolic pathways. A cross-lagged regression model was used to determine whether BMI at various ages during childhood predicted later cortisol production and metabolism parameters, or vice versa. The cross-lagged regression coefficients showed that BMI positively predicted cortisol metabolite excretion (*p* = 0.03), and not vice versa (*p* = 0.33). In addition, BMI predicted the later balance of 11β-hydroxysteroid dehydrogenase (HSD) activities (*p* = 0.07), and not vice versa (*p* = 0.55). Finally, cytochrome P450 3A4 activity positively predicted later BMI (*p* = 0.01). Our study suggests that changes in BMI across the normal range predict alterations in HPA axis activity. Therefore, the alterations in HPA axis activity as observed in earlier studies among children with obesity may be a consequence rather than a cause of increased BMI.

## 1. Introduction 

Over the past few decades, rates of childhood obesity have increased. The hypothalamus–pituitary–adrenal (HPA) axis, with cortisol as the end product, may be involved in the pathophysiology of obesity. Cortisol is known to influence appetite regulation, energy homeostasis, and the sensitivity to other metabolic hormones [1]. Circulating cortisol level reflects the balance between cortisol production and cortisol metabolism. It is eliminated from the circulation by the A-ring reductases 5α- and 5β-reductase, and by cytochrome P450 3A4 (CYP3A4). The availability of cortisol in tissues is regulated by 11β-hydroxysteroid dehydrogenase (HSD) isoenzymes. 11β-HSD type 1 is mainly expressed in the liver and adipose tissue, where it regenerates cortisol from inert cortisone, and 11β-HSD type 2 catalyzes the reverse reaction in the kidneys [2,3]. The expression of 11β-HSD type 1 in adipose tissue results in an increased availability of bioactive glucocorticoids in that tissue. 

Childhood obesity is, analogous to adult obesity, characterized by increased cortisol production and by flattening of diurnal rhythmicity in the secretion of cortisol [4,5,6,7,8,9]. In both children and adults with obesity, indices of cortisol production are positively associated with features of metabolic syndrome, i.e., a cluster of cardiovascular risk factors including abdominal fat distribution, raised blood pressure, hyperglycemia, and dyslipidemia [4,6,10]. 

It is unclear whether alterations in HPA axis activity are a cause or a consequence of obesity. Some evidence suggests that these alterations may be a consequence, since it has been demonstrated among prepubertal children with obesity that cortisol and cortisone in serum obtained at 8 a.m. decreased in those who managed to lose weight [11], particularly in those with obesity and insulin resistance [4]. A longitudinal study of low-income children who were followed between 4 and 8 years of age showed that being overweight at age 4 was associated with decreases in morning cortisol level and cortisol reactivity during a social stress test at age 8 [12]. Previous studies examining associations between childhood obesity and HPA axis activity measured cortisol in serum, saliva or hair [11,12,13], representing the current level or, in the case of hair, the average level over a prolonged period. Measurement of cortisol metabolites in urine enables an assessment of cortisol metabolism pathways.

There is a paucity of data on the temporal relationship between body mass index (BMI) within the normal range and cortisol metabolite excretion. Such information is essential, since in obesity multiple mechanisms may confound longitudinal relations between BMI and cortisol parameters. Relevant mechanisms are related to the degree of insulin resistance, elevation of blood pressure, and systemic inflammation. 

The aim of this study is to explore the complex relationship between cortisol metabolite excretion rate and BMI across adolescence. We hypothesized that cortisol metabolite excretion rate is dependent on BMI, implying that alterations in HPA axis activity are a consequence rather than a cause of obesity. 

## 2. Materials and Methods

### 2.1. Participants

We conducted a prospective follow-up study in healthy mono- and dizygotic twin pairs at ages 9, 12, and 17 years of age, as described previously [14]. Participants in this study were recruited from the Netherlands Twin Register (NTR), a population-based register [15,16]. They were invited to take part in the BrainScale project, which is a developmental study of cognition, hormones, and brain structure and function [17,18]. The BrainScale study is a collaborative project of the NTR at the Vrije Universiteit Amsterdam and the University Medical Center (UMC) Utrecht. In total, 109 families consented to participate, hence 218 subjects (109 twin pairs) were included in this study. Ethical approval was obtained from the medical ethics committee of the Amsterdam UMC, location VUmc (2014.023). Written informed consent was obtained from parents of the participants, and from the adolescents themselves at the third measurement around age 17 year. 

### 2.2. Study Protocol

At the ages of 9, 12 and 17 years, participants visited the study site at the Vrije Universiteit and UMC Utrecht for different tests. In the week prior to the study visit, urine samples were collected upon awakening in specially provided tubes. Participants or their parents were requested to store the tubes in their refrigerator and to bring them to the study visit. Samples were subsequently stored at −20 and −80 degrees Celsius, and thawed only once just before analysis. 

Body weight and height were measured by a researcher. Participants were measured barefoot, while wearing underclothing only. Height was measured with a fixed stadiometer, and weight with a balance scale. BMI was calculated and converted to standard deviation score, thereby adjusting BMI for age and sex according to Dutch norms [19]. 

### 2.3. Laboratory Analysis 

Analysis of cortisol and cortisol metabolites was conducted at the Edinburgh Clinical Research Facility Mass Spectrometry Core Laboratory. Glucocorticoid metabolites were measured by gas chromatography-tandem mass spectrometry (GC-MS/MS) [20]. Samples were analyzed in fifteen batches. Ratios of cortisol metabolites, representing the activities of various enzymes involved in cortisol metabolism, were calculated according to the indices in Table 1. The sum of cortisol metabolites represents cortisol production, and the other indices represent the various cortisol metabolism pathways [21].

### 2.4. Statistical Analysis

In line with previous analyses in this sample, extreme outliers are defined as >3SD above the phenotypic mean or twin pairs with highly discordant outcomes; on average, six and one per index, respectively, were excluded from the statistical analysis [14,21]. Next, the data were corrected for batch effects by fitting a random effects model, in which batch was treated as a random effect [22]. 

### 2.5. Statistical Modeling 

We used a cross-lagged regression model to determine whether BMI at various ages across adolescence predicted later cortisol parameters and vice versa (Figure 1). 

As shown in Figure 1, the model includes autoregressions, cross-lagged regression, and synchronous correlations. The autoregressions involved regressing BMI at age 9 on BMI at age 12, and BMI at age 12 on BMI at age 17, i.e., a regression on the same trait measured at an earlier age. The same applies to the autoregressive relation linking the cortisol indices at the ages of 9, 12, and 17. In Figure 1, the parameters bxx1 and bxx2 are the autoregressive coefficients of BMI, and the parameters byy1 and byy2 are the autoregressive coefficients of the cortisol indices. To assess the direction of the relationships, we included cross-lagged regression, i.e., regression on a different trait at an earlier age. For example, the cortisol index at age 12 (y12) was regressed on BMI at age 9 (parameter bxy1), and BMI at age 12 was regressed on the cortisol index at age 9 (parameter byx1). These cross-lagged regressions were included at ages 12 and 17. Finally, as shown in Figure 1, synchronous, i.e., within age, correlations were included between BMI and the phenotype at age 9 (ryx1), and between the residuals of BMI and the cortisol index at ages 12 and 17 (rxy2 and rxy3). 

To test the direction of the causal effect, we tested the hypothesis bxy1 = bxy2 = 0, where the direct effect of BMI on the cortisol index is zero. Subsequently, we tested the hypothesis byx1 = byx2 = 0, where the direct effect of the cortisol index on BMI is zero. Both tests were two likelihood ratio tests with two degrees of freedom and involved comparing the likelihood of the full model with the likelihood of the constrained model. Given the relatively small sample size, we adopted an alpha of 0.10 in conducting the tests. If there is no direct effect of BMI on the cortisol index, the first test (hypothesis bxy1 = bxy2 = 0) is expected to be not significant. If the test is significant with a *p*-value below 0.10, we conclude that there is a direct effect. The same applies to the test of the direct effect of the cortisol index on BMI (hypothesis byx1 = byx2 = 0). If both tests are significant, we would infer a reciprocal causal effect of BMI on the cortisol index, and vice versa. In fitting these models and carrying out the tests, we included sex as a fixed regressor. We consistently used robust maximum likelihood estimation. As we treated each person as an individual case, we correct the standard errors for clustering (i.e., of twins in families). A sample size calculation concerning statistical power was not performed prior to analysis, as the original study was designed to analyze associations between cognition, hormones, and brain development and not specifically to address temporal relations between cortisol parameters and BMI. 

## 3. Results

A total of 218 participants, of whom 50% were male, were included in this study. They were tested at 9.1 (± 0.1), 12.2 (± 0.3), and 17.2 (± 0.2) years of age. In total, 542 samples were analyzed, of which 213, 167, and 162 were obtained at the ages of 9, 12, and 17 years, respectively. Their characteristics are described in Table 2. The mean (± SD) standard deviation score for body mass index (weight(kg)/height(m)^2^) was 0.14 (± 0.93), 0.45 (± 1.00), and 0.27 (± 1.08) at the ages of 9, 12, and 17 years, respectively.

### Association between Body Mass Index and Cortisol Parameters

Figure 2 displays the standardized cross-lagged regression coefficients for cortisol metabolite excretion rate. The coefficients associated with BMI were large: 9–12 years = 0.84 and 12–17 years = 0.74, indicative of a relatively high stability of BMI. The coefficients associated with cortisol metabolite excretion rate were low: 9–12 years = 0.20 and 12–17 years = 0.07. The synchronous correlations between (residuals of) BMI and cortisol metabolite excretion rate were 0.18 (rxy1), 0.04 (rxy2) and −0.01 (rxy3) at the ages of 9, 12, and 17 years, respectively. The cross-lagged regression coefficients bxy1 and bxy2, representing the influences of cortisol metabolite excretion rate at age 9 on BMI at age 12, and of cortisol metabolite excretion rate at age 12 on BMI at age 17, equaled 0.12; 90% CI 0.00–0.23 and 0.15; 90% CI 0.01–0.29, respectively, and the hypothesis bxy1 = bxy2 = 0 was rejected based on a likelihood ratio test statistic (*p* = 0.03). In contrast, the test of the cross-lagged regression coefficients byx1: 0.08, 90% CI −0.00–0.15 and byx2: 0.02, 90% CI −0.08–0.13, representing the influence of cortisol metabolite excretion rate on later BMI, was not significant (*p* = 0.33). 

Table 3 displays the cross-lagged regression coefficients for all cortisol parameters. In addition to influences of BMI on later cortisol metabolite excretion rate, we found that BMI predicted the later balance of 11β-HSD activities (*p* = 0.07), although the direction of association was different for the two periods. No other associations were found.

Among the comparisons addressing the impact of cortisol parameters on later BMI, only cytochrome P450 3A4 activity was significant (*p* = 0.01). This could be attributed to a high correlation between the ages of 12 and 17 years: 0.16, 90% CI 0.07–0.24, but not between 9 and 12 years of age: 0.02, 90% CI −0.05–0.09.

## 4. Discussion

In this longitudinal study in healthy twins, we aimed to elucidate the temporal relationship between cortisol parameters and BMI across adolescence, by investigating the directionality of associations in a cross-lagged regression model. The most important finding from our study was that BMI positively predicts cortisol metabolite excretion at the next assessment, but not vice versa, suggesting that greater cortisol production is a consequence rather than a cause of increases in childhood BMI.

Our results are supported by findings from others, suggesting that increases in HPA axis activity are a consequence rather than a cause of childhood obesity. More specifically, it has been demonstrated among children with obesity that serum levels of adrenocortical hormones decreased in those who managed to lose weight [4]. Others found that being overweight or obese at preschool predicted reductions in morning cortisol and cortisol reactivity during a social stress test in middle childhood [12]. Our study adds that such associations can be extrapolated to the normal BMI range. 

This study suggests that not only cortisol production, but also cortisol metabolism, is influenced by previous BMI. Since the availability of cortisol in tissues is regulated by 11β-HSD isoenzymes, dysregulation in the expression of these enzymes, specifically in adipocytes versus hepatic tissue, has potential implications for HPA axis activity. This study suggests that BMI predicts the balance between these two isoenzymes at a later age. Given the lack of association between BMI and 11β-HSD type 2 activity, our data suggest that BMI predicts later 11β-HSD type 1 activity, although for unknown reasons the direction of association was negative in the first period and positive in the second period. Greater 11β-HSD type 1 activity implies a higher exposure to bioactive glucocorticoids. In adults, inconsistencies in the relationship between BMI and 11β-HSD type 1 activity have also been observed [23].

Contrary to our expectations, we found a positive correlation between CYP3A4 activity, which functions to eliminate cortisol rapidly from the circulation, and later BMI. Although CYP3A4 activity is known to eliminate only a small proportion of circulating cortisol, the direction of association suggests that this may be a mechanism that protects against corticosteroid-induced BMI gains.

Our study has several strengths and limitations. One major strength was the long follow-up period of 8 years. The follow-up rate was relatively high for an age group that is notoriously difficult to engage in longitudinal studies. The longitudinal design of our study allowed us to elucidate the temporal relationship between cortisol parameters and BMI. Cross-lagged path models may suggest causality when a randomized control trial is not possible. Another strength was that participants were recruited from a population-based twin register and were therefore representative of the general Dutch population. However, the BMI of our sample was slightly higher than the Dutch population norm. Our study also has its limitations. Enzymatic activities were estimated by ratios of corticosteroid metabolites in morning urine, which estimate activity only globally. Another limitation was the lack of information on body composition estimates, such as Dual Energy X-ray Absorptiometryand bioelectrical impedance, and body fat distribution, since alterations in HPA axis activity have traditionally been linked to intra-abdominal adipose tissue accumulation [24]. 

## 5. Conclusions

Our study suggests that the alterations in HPA axis activity accompanying obesity precede BMI gains. The complex temporal relation between BMI and later 11β-HSD type 1 activity might be dependent on age-related changes in fat distribution, requiring further study. Our observation that CYP3A4 activity positively predicted later BMI warrants further study. 

## Figures and Tables

**Figure 1 nutrients-12-01525-f001:**
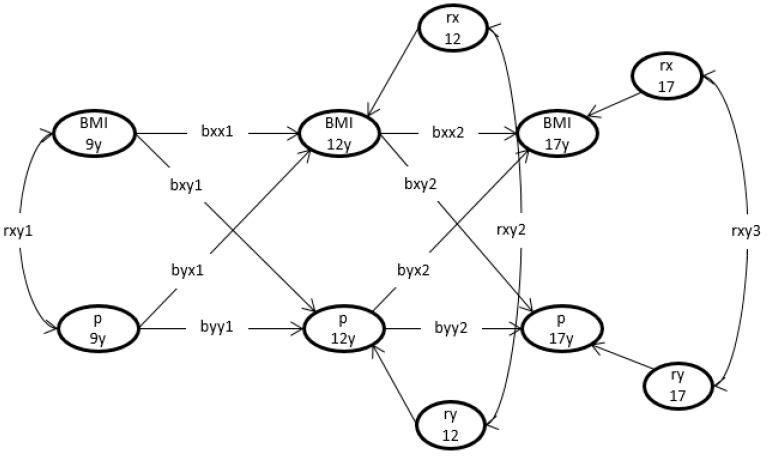
Representation of the model to explore the temporal relation between BMI and cortisol indices. The arrows represent the causal pathways, and the double-headed arrows represent the correlations. BMI = body mass index, assessed at ages 9, 12, and 17 years. P represents the cortisol parameters at those ages. This model includes autoregressions, cross-lagged regressions, and synchronous correlations. The parameters bxx1 and bxx2 are the autoregressive coefficients of BMI, which link the successive BMI values. The parameters byy1 and byy2 are the autoregressive coefficients of the cortisol indices, which link the successive cortisol indices. The parameters bxy1 and bxy2 are the cross-lagged regression coefficients in the regression of the cortisol index on BMI, and byx1 and byx2 are the cross-lagged regression coefficients in the regression of BMI on the cortisol index. Finally, rxy1, rxy2, and rxy3 are the synchronous correlations between BMI and the cortisol index at age 9 (rxy1), and between the residuals of BMI and the cortisol indices at ages 12 and 17 (rxy2 and rxy3, respectively).

**Figure 2 nutrients-12-01525-f002:**
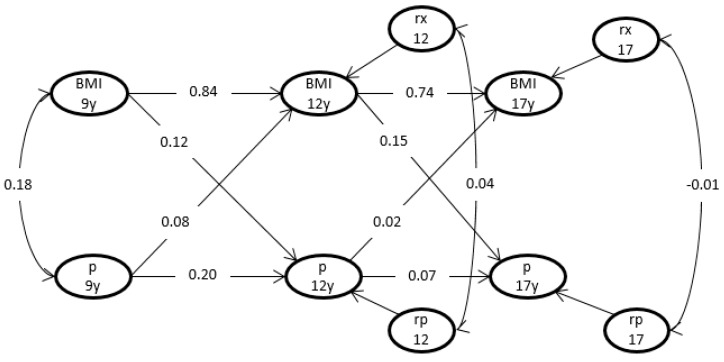
Estimates for parameters in a model for BMI and cortisol metabolite excretion rate. Cross-lagged correlation coefficients between BMI and cortisol metabolite excretion rate at ages 9, 12, and 17 year. BMI 9y = body mass index at 9 years of age. p 9y = cortisol metabolite excretion rate at 9 years of age. rx and rp are residuals for BMI and cortisol metabolite excretion rate at ages 12 and 17 years.

**Table 1 nutrients-12-01525-t001:** Cortisol parameters reflecting cortisol production and cortisol metabolism pathways.

Parameter	Index
(THF + allo-THF + THE + α-cortol + β-cortol + α-cortolone + β-cortolone)/creatinine	Sum of cortisol metabolites (cortisol metabolite excretion rate)
allo-THF/F	5α-reductase activity
THF/F	5β-reductase activity (a)
THE/E	5β-reductase activity (b)
F/E	11β-HSD type 2 activity
(THF + allo-THF)/THE	Balance of 11β-HSD activities
6β-OH cortisol/F	Cytochrome P450 3A4 activity

Abbreviations: THF, tetrahydrocortisol; THE, tetrahydrocortisone; HSD, hydroxysteroid dehydrogenase; F, cortisol; E, cortisone.

**Table 2 nutrients-12-01525-t002:** Characteristics of participants.

		9 years (*n* = 207)	12 years (*n* = 175)	17 years (*n* = 174)
Height	cm	138.7 ± 5.2	152.3 ± 7.1	173.6 ± 8.2
	SDS	0.04 ± 0.82	−0.56± 0.98	−0.16 ± 0.88
Weight	kg	31.4 ± 4.5	43.0 ± 9.0	64.0 ± 9.6
	SDS	0.31 ± 0.93	0.23± 0.99	0.26 ± 1.06
Body mass index	kg/m^2^	16.3 ± 1.7	18.7 ± 2.5	21.3 ± 3.0
	SDS	0.14 ± 0.93	0.45 ± 1.00	0.27 ± 1.08

Values represent mean ± SD.

**Table 3 nutrients-12-01525-t003:** The relation between BMI and cortisol indices.

Index	bxy1	bxy2	*P*-Value	byx1	byx2	*P*-Value
Sum of cortisol metabolites (cortisol metabolite excretion rate)	0.12 (0.00–0.23)	0.15 (0.01–0.29)	0.03	0.08 (−0.00–0.15)	0.02 (−0.08–0.13)	0.33
5α-reductase activity	0.02 (−0.12–0.16)	−0.13 (−0.26–0.00)	0.30	0.03 (−0.04–0.11)	0.05 (−0.07–0.17)	0.57
5β-reductase activity (a)	0.01 (−0.09–0.11)	0.09 (−0.03–0.21)	0.42	0.00 (−0.05–0.06)	0.09 (0.01–0.17)	0.15
5β-reductase activity (b)	0.08 (−0.06–0.21)	−0.11 (−0.26–0.04)	0.33	0.07 (−0.01–0.14)	0.04 (−0.04–0.13)	0.21
11β-HSD type 2 activity	0.08 (−0.05–0.20)	−0.04 (−0.20–0.13)	0.61	−0.01 (−0.07–0.05)	−0.04 (−0.12–0.04)	0.65
Balance of 11β-HSD activities	−0.10 (−0.19–0.01)	0.12 (−0.01–0.25)	0.07	0.03 (−0.06–0.13)	0.06 (−0.03–0.15)	0.55
Cytochrome P450 3A4 activity	−0.08 (−0.24–0.07)	0.01 (−0.16–0.18)	0.65	0.02 (−0.05–0.09)	0.16 (0.07–0.24)	0.01

This table includes the cross-lagged regression coefficients along with their 90% confidence intervals. bxy1: cortisol parameter at age 12 year regressed on BMI at age 9 year, bxy2: cortisol parameter at age 17 year regressed on BMI at age 12 year, byx1: BMI at age 12 year regressed on cortisol parameter at age 9 year, byx2: BMI at age 17 year regressed on cortisol parameter at age 12 year. The *p*-value is based on the 2 degrees of freedom test of bxy1 = bxy2 = 0 and byx1 = bxy2 = 0. Bold: *p*-value < 0.10, we conclude that the parameters are not zero.

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
