# Peer review of "Exploring the Temporal Relation between Body Mass Index and Corticosteroid Metabolite Excretion in Childhood"

_nutrients, 2020, doi:10.3390/nu12051525_

Round 1

Reviewer 1 Report

Dear authors

Generally:

  • You must avoid the frequent use of parenthesis.
  • There is a piece of information in the abstract, that I did not notice in discussion ex. Cytochrome P450 3A4.
  • The reference "21" is not published yet. 

It is necessary to alter some parts of the manuscript because there is unclear meaning or it is necessary to put some of the study's details and descriptions in different manuscripts spots.

Lines:

  • 49 to 54
  • 59 rewrite the aim of the study without the participants' details. These details be a part of the "methodology"
  • 61 It is better the study's hypothesis to be a part of the "methodology"
  • 64 It is necessary to present the number of participants in "methodology" clearer
  • 72 - 82 I do not understand where the study conducted 
  • 92 Put the table's details under it.
  • 94 The participants' exclusions from the study must be a part of the study's' "participants" and not of the statistic.
  • 100 The aim of the study must be deleted from the "statistic".
  • 135 This sentence is contradictory to the study's title.
  • 137 Delete the bibliography and the discussion from the "results" 

Reviewer 2 Report

This objective of this paper is an attempt to determine if BMI predicts changes in HPA-axis activity (as measured by corticosteroid metabolites) over time. This paper would be one of few which examine this temporal relationship and challenges the traditional model which proposes that HPA-axis dysfunction contributes to obesity. The authors’ data seem to suggest that BMI predicts changes in HPA-axis activity (as defined by corticosteroid metabolites). Although the findings do appear to be very intriguing, the authors require a lot more background to introduce the corticosteroid metabolites and to better translate what this means in comparison to other HPA literature that has focused on cortisol concentration in saliva, blood, hair. There is not enough information overall for this paper to appeal to those in the obesity field or epidemiologists who may not fully understand cortisol mechanisms. The paper is short and full of jargon and needs a lot of clarity.

Abstract

Results in lines 27-28: What does this statement mean?  “BMI predicted the later balance of 11-beta-HDS activities?” Is that result good or bad?

Result in 29-30: Was high or low Cytochrome P450 3A4 activity that predicted High/Low BMI? The sentence in Line 29-30 “Our study suggests that the alterations in HPA axis activity accompanying obesity follow after BMI gain” is awkward. Please clarify.

Introduction
To my knowledge, this method of assessing HPA-axis activity is not as commonplace as other methods (serum cortisol, diurnal cortisol patterns from saliva) therefore the authors much provide more literature that shows who these metabolites could associate with obesity or disease. There is not a strong link shown between the basic biology of HPA axis metabolism and the epidemiological evidence that shows it linked to obesity or disease risk, so it is the author’s duty to make this case more clearly for the reader. If this new to the field, then the authors state that this is new and make a case for the novelty of the paper.

Line 61-62: The authors hypothesize that cortisol production and metabolism are dependent on BMI, however I believe it may be more correct (or clearer) to say “cortisol production and excretion” or just cortisol metabolism. This comes back to the lack clarity in explaining the intricacies of cortisol metabolism.

What do authors mean to say with the statement “increased HPA activity”?Authors should clearly follow-up with the specific direction of hypothesized association for each metabolite would make this paper much clearer. If it unknown, then state that this exploratory and it is unknown.

Methods

  • Line 80: Is a “standard deviation score” meaning a z-score? Is this adjusted to age and sex? I presume it is however I could not figure it out based on the citation. Please fix the reference. I cannot tell which curve the authors utilized to calculate BMI SDS. Also, on a related note, Table 2 which describes length (please say height instead), weight and BMI doesn’t appear to reflect true z-scores. The values are too high, but it could also be that the SDS method is not clearly explained. This is very concerning and must be clarified since BMI in children must be adjusted.
  • A line-by-line explanation of the parameters in Table 1 is necessary. What does each one of these metabolites represent with relationship to cortisol metabolism? Only the first parameter has an explanation so it would be useful if all parameters were explained.

Results

  • Why are no descriptive statistics shown for variables in Table 1? Line 86 states those values were calculated but they are not shown. Why?
  • I am unable to properly comment on the statistical techniques since I am unfamiliar with methods.

Discussion

  • The discussion is very sparse- needs a lot of improvement. I will provide a few suggestions: 1) The 3rdparagraph contains information that could be introduced earlier in the paper (the introduction).
  • Lines 181-184: But how do the authors know this? In other words, what were the exact findings that led to this conclusion?
  • Please consolidate the strengths of the study and limitations (do not alternate, back and forth).
  • The sentence in lines 217-218 is not sufficient. In the obesity field, BMI is only proxy for body composition, therefore the authors must state that a limitation is the lack of precise body composition measures (such as DEXA, BodPod or even bioelectrical impedance). Related to the point above, cortisol (HPA axis activity) has been traditionally linked to abdominal adiposity. This study lacks waist measurements or MRI imaging of visceral fat. Please discuss all of these relevant points.
  • The conclusion omits the finding on CYP3A4. Although this finding contradicts seems to contradict one of the authors’ hypothesis (as stated in line 202), it also warrants more study.

Round 2

Reviewer 1 Report

The authors proceeded to the changes that the manuscript needed.